# Random Shuffling Data for Hyperspectral Image Classification with Siamese and Knowledge Distillation Network

**Zhen Yang [1,2], Ying Cao [1], Xin Zhou [1,*], Junya Liu [1], Tao Zhang [3] and Jinsheng Ji [4]**

1. The School of Information and Electromechanical Engineering, Jiangxi Science and Technology Normal University, Nanchang 330013, China; yangzhen@jxstnu.edu.cn (Z.Y.); 1020230302@jxstnu.edu.cn (Y.C.); 1020150923@jxstnu.edu (J.L.)
2. Key Laboratory of System Control and Information Processing, Ministry of Education, Shanghai 200240, China
3. Shanghai Key Laboratory of Intelligent Sensing and Recognition, Shanghai Jiao Tong University, Shanghai 200240, China; sjtu--zt@sjtu.edu.cn
4. School of Electrical and Electronic Engineering, Nanyang Technological University, Singapore 639798, Singapore; jinsheng.ji@ntu.edu.sg
* Correspondence: zhouxin@jxstnu.edu.cn; Tel.: +86-15797682020

**Abstract:** Hyperspectral images (HSIs) are characterized by hundreds of spectral bands. The goal of HSI is to associate the pixel with a corresponding category label by analyzing subtle differences in the spectrum. Due to their excellent local context modeling capabilities, Convolutional Neural Network (CNN)-based methods are often adopted to complete the classification task. To verify whether the patch-data-based CNN methods depend on the homogeneity of patch data during the training process in HSI classification, we designed a random shuffling strategy to disrupt the data homogeneity of the patch data, which is randomly assigning the pixels from the original dataset to other positions to form a new dataset. Based on this random shuffling strategy, we propose a sub-branch to extract features on the reconstructed dataset and fuse the loss rates (RFL). The loss rate calculated by RFL in the new patch data is cross combined with the loss value calculated by another sub-branch in the original patch data. Moreover, we construct a new hyperspectral classification network based on the Siamese and Knowledge Distillation Network (SKDN) that can improve the classification accuracy on randomly shuffled data. In addition, RFL is introduced into the original model for hyperspectral classification tasks in the original dataset. The experimental results show that the improved model is also better than the original model, which indicates that RFL is effective and feasible. Experiments on four real-world datasets show that, as the proportion of randomly shuffling data increases, the latest patch-data-based CNN methods cannot extract more abundant local contextual information for HSI classification, while the proposed sub-branch RFL can alleviate this problem and improve the network's recognition ability.

**Keywords:** patch-data-based CNN method; local contextual information; random shuffling strategy; network SKDN consists of two sub-branches; HSI classification

## 1. Introduction

Compared to traditional two-dimensional RGB images, hyperspectral images (HSIs) have many continuous spectral bands. Therefore, they also contain more information that allows us to better perform tasks, such as object detection, change detection, and geological exploration. Land cover classification is one of the most important issues within these fields, as it aims to apply specific semantic labels to the pixels of the entire HSI on the basis of the unique spatial-spectral features of the HSI. In recent years, there has been increasing interest in Deep Learning (DL) methods for solving the problem of HSI classification. Depending on the shape of the training data, we can divide these methods into the following two categories: the single-pixel-based methods and the patch-data-based methods.

Existing DL models often ignore the inherent relationship between pixels in patch data because they are designed for Euclidean data. In recent years, Graph Convolutional Networks (GCNs) have received increasing attention as a representative of the single-pixel-based methods due to their ability to perform convolutions on arbitrarily structured graphs. By encoding HSI into a graph, the correlation between adjacent land cover can be explicitly exploited, and the spatial context structure of HSI can be better modeled by GCNs [1]. Shahraki et al. [2] proposed a model combining 1-D Convolutional Neural Networks and GCNs for HSI classification. However, due to the large number of pixels in HSIs, using each pixel as a node of a graph has been shown to incur huge computational costs and limit its applicability. To solve this issue, Hong et al. [3] developed a new model named mini-GCN, which can be used to train large GCNs in a mini-batch process. Liu et al. [1] proposed a heterogeneous model called CEGCN, in which CNN and GCN sub-networks generate complementary feature information at pixel and super-pixel levels. To explore Non-Euclidean structures and reduce high computational costs, Bai et al. [4] developed a graph attention model with an adaptive graph structure mining approach (GAT-AGSM). Liu et al. [5] designed a multi-level network based on U-Net operating on a superpixel structured graph, named MSSGU, to learn multi-level features on multi-level graphs and overcome the limitation of specific superpixel segmentation in modeling.

Generally, the methods mentioned above mainly belong to single-pixel-based methods. Recently, researchers still pay more attention to the patch-data-based methods, which can extract more discriminative information and deeper features in model training. In particular, Convolutional Neural Network (CNN) is very popular among patch-data-based methods. Since the feature information contained in a single pixel is limited, CNN cannot extract local context information from the data. To solve this problem, researchers changed the shape of the input data. Specifically, the pixel is placed in the center of the patch data. The pixels around the classified pixel are also included in the patch data. This allows the model to extract more local contextual information from the patch data to enhance the classification accuracy in the central pixel.

In the early days of 2016, researchers mainly focused on HSI classification based on 1D-CNNs and 2D-CNNs [6]. Chen et al. [7] proposed a model that uses Stacked Autoencoders (SAE) for high-level information extraction and 1D-CNN for low-level information extraction. In addition to methods based on 1D-CNNs, 2D-CNN-based methods are often introduced into the HSI network to extract spatial feature information. Shen et al. [8] proposed a model called ENL-FCN to incorporate the long-range context information, which is constructed by a deep fully convolutional network and an efficient non-local module. Since residual blocks have been extensively adopted in the HSI network, Zhong et al. [9] developed a model to improve the accuracy of HIS classification, which is named the Spectral-Spatial Residual Network (SSRN), and greatly enhanced the feature utilization rate by using front-layer feature information to complement back-layer features. Furthermore, Paoletti et al. [10] proposed a model named DPRN that also adopted the residual module. Inspired by the 3D convolution, 3D convolution blocks have been widely applied to HSI networks, which utilize spectral-spatial information to achieve more satisfactory classification results. In [11], Zhong et al. developed 3D Deep-Residual Networks (3D-ResNets) to reduce the influence of model size. To mine deeper spectral-spatial feature information from HSIs, Zhang et al. [12] also used 3D-CNNs to construct a 3D-DenseNet model.

In recent years, combining 2D and 3D convolutions and optimizing the structure of CNN-based models have gradually become a hotspot. Combining the advantage of 2D-CNN and 3D-CNN, some researchers designed a dual-branch structure to enhance the classification accuracy of HSI. For instance, Zheng et al. [13] proposed a mixed CNN with covariance pooling, named MCNN-CP, where covariance pooling is used to mine second-order information from the spectral-spatial feature. Roy et al. [14] developed a model that is a 3D convolution block combined with a 2D convolution block, which is called a hybrid spectral CNN (HybridSN). By introducing an efficient residual structure, the network parameters can be optimized, and a lightweight design can reduce the computational

complexity of the model. Wu et al. [15] designed a re-parametrized network, abbreviated as RepSSRN, which reparametrizes the Spectral-Spatial Residual Network (SSRN). Inspired by the Dense Convolutional Network, Wang et al. [16] proposed a fast dense spectral-spatial convolution (FDSSC) algorithm. In another work [17], a network named CMR-CNN adopted the 3D residual blocks followed by the 2D residual blocks together to capture the spatial-spectral feature information of the HSI. To reduce the network parameters and computational complexity, Meng et al. [18] proposed a lightweight spectral-spatial convolution HSI classification module (LS2CM), and Li et al. [19] designed a lightweight network architecture (LiteDenseNet). The computational complexity and network parameters in their work are much lower than counter-intuitive deep learning methods. Although these methods mentioned above are effective in improving the classification performance of HSI, it is difficult to overcome the issue caused by the limited training samples and increasing network layers. So, much work has been done to introduce the attention mechanism and transformer models into the HSI network. Sun et al. [20] constructed a model called the spectral-spatial attention network (SSAN) to acquire important spectral-spatial information in the attention regions of patch data. Li et al. [21] proposed a model called DBDA to capture a variety of spectral-spatial information, which is a dual-branch network with two attention mechanisms. He et al. [22] designed a model called the Spatial-Spectral Transformer (SST), which extracted spatial features via the VGGNet network and established the relationship between adjacent spectra by using the dense transformer blocks. Similarly, Sun et al. [23] designed a network that adopts a Gaussian weighted feature tokenizer to capture high-level semantic features, which is called SSFTT. By improving the traditional transformer model, Hong et al. [24] designed a novel method that is widely used in HSI classification tasks, which is called SpectralFormer (SF). Although these networks are excellent at capturing spectral signatures, these models cannot capture the local contextual information of patch data well, and they make insufficient use of the spatial features in HSI.

Besides the above limitations, the patch-data-based methods also tend to rely on the local neighborhood information of the patch data in the training process, and sometimes, overfitting is caused by improper setting of the training proportion. To verify whether the patch-data-based CNN methods depend on the homogeneity of patch data during the training process and evaluate the ability of the HSI method to extract spatial location information, we rethink the HSI classification process from the data perspective in the patch-data-based method and design a novel strategy to reconstruct the original dataset. Based on this strategy, we also propose a new model based on the Siamese and Knowledge Distillation Network (SKDN) to complete the classification task. The most important contributions of this work can be summarized in the following way.

1. We use the proposed strategy of randomly shuffling data to explore the influence of patch data homogeneity features in HSI classification networks. Specifically, this strategy involves randomly assigning the pixels in the original dataset to other locations to construct a new dataset. Therefore, the new patch data after the random strategy contains richer information about species categories;

2. We propose a sub-branch to extract features from the reconstructed dataset and fuse the loss values (RFL), which uses a designed loss function in RFL to compute and fuse loss values from two sub-branches. The novelty of this loss function is that the loss rate computed by RFL in the new data is cross combined with the loss rate calculated in the original data from another sub-branch to further optimize the network. Thus, the proposed network can not only enhance the recognition ability of the model, but also increase the classification accuracy on randomly shuffled data, which is based on the Siamese and Knowledge Distillation Network (SKDN) and is constructed from two sub-branches;

3. We also introduce the proposed sub-branch RFL into the original network to further explore the effectiveness of the RFL, and we let the original model and its improved model achieve HSI classification on the original dataset. The experiments show that

the classification performance of the improved model is better than that of the original model, so it can also prove that the proposed sub-branch is effective and feasible;

4. Experiments conducted on several typical datasets show that, as the proportion of randomly shuffled data increases, the latest patch-data-based CNN methods are unable to extract more abundant local contextual information for HSI classification, while the proposed sub-branch can alleviate this problem.

The rest of this paper consists of the following sections: Section 2 presents the methodology, Section 3 presents and analyzes the experiments, Section 4 discusses the usefulness of the proposed methods, and Section 5 is the work's conclusion.

## 2. Methodology

More details of the developed random shuffling strategy and the new network SKDN constructed by the proposed sub-branch will be introduced and discussed carefully in this section.

### 2.1. Randomly Shuffling the Pixels

It is well known that most pixels in the patch data belong to the same category in the CNN-based networks in HSI classification. CNNs have shown that they can capture spectral-spatial feature information and local contextual information in training patch data. They largely rely on the data homogeneity of the patch data in the model's training, leading to overfitting if the value of the training ratio is not chosen correctly. Here, we designed a random shuffling strategy to disrupt the data homogeneity of the patch data, which is randomly assigning the pixels from the original dataset to other positions to form a new dataset. The random shuffling of the pixel scheme can be seen in Figure 1. It consists of reconstructing the dataset by randomly shuffling the position of pixels. The false color image of the rebuilt dataset based on Indian Pines with various random shuffling ratios is displayed in Figure 2. We found that as the proportion of random shuffling increased, so did the category differences in each pixel cube. This random shuffling strategy can increase the diversity of extracted features from the new dataset. Specifically, for each new neighboring cube that contains many types of categories, CNNs can learn more category information about different ground objects and extract more local context information during feature extraction. We also present hereafter the advantage of the proposed random shuffling strategy over the traditional training dataset in HSI classification. In general, the 3D-conv operations are adopted to acquire more spectral-spatial features in HSI, while the 2D-con operations pay more attention to acquiring more spatial features. If most of the training samples in the neighboring cube belong to the same category, the model can easily lead to ignoring some category information about other categories, resulting in a large reliance on the data homogeneity of the patch data in the training process. By randomly rearranging the pixel positions, the patch data of HSI can be reconstructed. This not only serves to test the robustness of the network, but also helps the CNN-based model extract more local context information and category information in the patch data. This strategy also increases the training difficulty of the CNN-based network in the training process. Therefore, in comparison to the traditional method of training patch data, the proposed strategy enables us to rethink the task of HSI classification from the perspective of patch data with feature extraction.

### 2.2. Hyperspectral Image Classification Network SKDN

By giving the traditional Siamese network a classification structure, it can have the ability to classify and additionally calculates a cross-entropy loss for each individual sample. Inspired by this method, we construct our network for HSI classification, with the difference that we only compute the cross-entropy loss in our structure. The Knowledge Distillation (KD) network is designed to transfer the extracted knowledge from a larger model into a smaller network for knowledge preserving and computationally inexpensive deep models [25]. Based on this theory, we design two branches to construct our model,

where one branch extracts features from the original dataset as a teacher network, and the other branch extracts features from the reconstructed dataset as a student network. The difference is that we design a loss function to merge the loss rates of the two branches. Although the neighborhood of the central pixel in the reconstructed dataset contains a variety of species, it also makes it more difficult for the model to classifying central pixels of patch data. Therefore, we propose a network SKDN based on the input form of the Siamese network and the distillation mode of the KD network to improve the classification ability. The structure of the network SKDN is illustrated in Figure 3. It consists of two sub-branches, which are used to extract deep features and local context information in the patch data from the original dataset and the reconstructed dataset, respectively. The weights are not shared between the backbones used to extract features in the two branches. Finally, the proposed new loss function is used to fuse the loss rates of the two branches in order to improve the classification effect of the model.

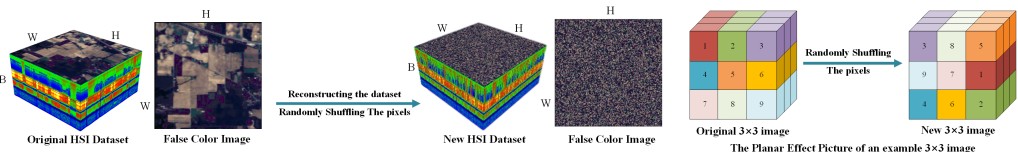

**Figure 1.** The use of random shuffling strategy to reconstruct the HSI dataset. H, W, and B, respectively, represent the length of HSI, the width of HSI, and the spectral number of HSI. It is randomly assigning the pixels from the original dataset to other positions to form a new dataset. We can that see the False Color Image of the new dataset is different from the False Color Image of the original dataset. The Planar Effect Picture shows the strategy applied to a sample $3 \times 3$ image. Note that the position of the spectrum corresponding to the shuffled pixel also changes as its pixel position changes.

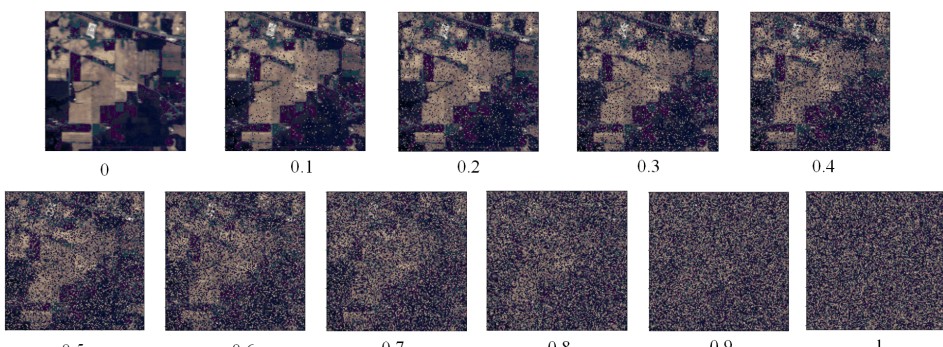

**Figure 2.** The False Color Image of the reconstructed dataset based on Indian Pines with different random shuffling ratios.

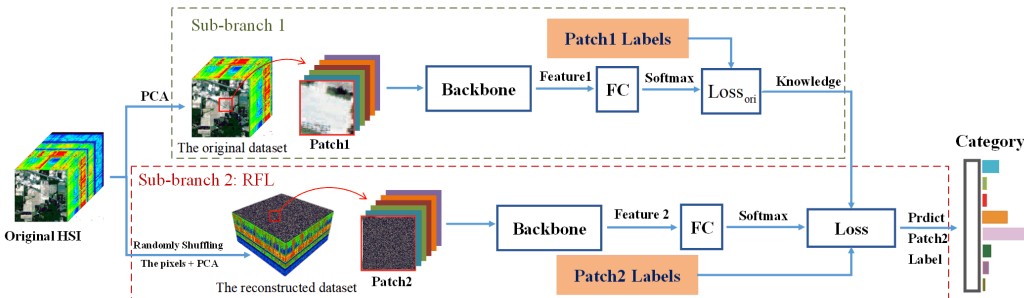

**Figure 3.** The architecture of the developed network SKDN, which consists of two sub-branches. Sub-branch 1 extracts the information and computes the loss value as the knowledge of the original dataset. Indeed, the sub-branch RFL extracts the feature and computes the loss value of the reconstructed dataset, then fuses the knowledge in the new loss function Loss.

**Differentiated input strategy.** Based on the random shuffling strategy, we also design a different input strategy that feeds patch data from different datasets into the dual branches of our network SKDN, providing more local context information to the model, while most existing methods use the original dataset. To be specific, we are feeding two sets as the input patch data for two branches in SKDN: the data input of one branch from the original patch data, and the other one from the reconstructed patch data. So, the input data in the network SKDN are different from the traditional Siamese networks, which often feed with different size input data from the same dataset for its model training. Since the neighborhood of the central pixel in the reconstructed dataset contains a variety of species, these patch data have richer species categories than the original, which means that they can provide more local context information to the model's training to strengthening the model's recognition ability. We also transfer the knowledge in the feature extraction backbones to improve the adaptability of the two sub-branches with different input data.

**Weighted Cross-Entropy Loss.** The sub-branch RFL of network SKDN includes two loss functions. The computational process of the loss function is given by

$$L = \alpha \times L_{ori} + \beta \times L_{rec}$$

$$L_{ori}\left(\hat{y}_n, y_n\right) = L_{rec}(\hat{y}_n, y_n) = -\frac{1}{N} \sum_{n=0}^{N-1} y_n \log(\hat{y}_n) \tag{1}$$

We use two weighted parameters ($\alpha$ and $\beta$) to fuse the two loss values learned from the dual branch to balance the differential information due to different inputs. Here, we also set the initial values of $\alpha$ and $\beta$ to 1 and 0.5, respectively. $y_n$ is the label of the input patch-data, and $\hat{y}_n$ is the predicted value of the network output result. The $L_{rec}$ represents the loss value calculated on the RFL, where the computational process is performed on the reconstructed dataset. The $L_{ori}$ represents the loss value calculated on sub-branch 1, where the computational process is performed on the original dataset. Note that the input patch data that are sent to the two sub-branches are different, so the two labels $y_n$ in $L_{ori}$ and $L_{rec}$ are also different. Furthermore, we still adopt a common Adam optimizer to optimize the network with L.

## 3. Experiments

To test the feasibility of the strategy, four real-world datasets from HSI were selected for the experiment in this paper. They are Indian Pines, Pavia University, Salinas, and Kennedy Space Center. We can see more details of these public datasets in Table 1.

**Table 1.** Numbers of samples and land cover classes in four public datasets.

| | Indian Pines | | | Pavia University | | | Salinas | | | Kennedy Space Center | |
|---|---|---|---|---|---|---|---|---|---|---|---|
| ID | Class Name | Samples | ID | Class Name | Samples | ID | Class Name | Samples | ID | Class Name | Samples |
| 1 | Alfalfa | 46 | 1 | Asphalt | 6631 | 1 | Brocoli_green_weeds_1 | 2009 | 1 | Scurb | 761 |
| 2 | Corn-notill | 1428 | 2 | Meadows | 18,649 | 2 | Brocoli_green_weeds_2 | 23,726 | 2 | Willow swamp | 243 |
| 3 | Corn-mintill | 830 | 3 | Gravel | 2099 | 3 | Fallow | 1976 | 3 | CP hammock | 256 |
| 4 | Corn | 237 | 4 | Trees | 3064 | 4 | Fallow_rough_plow | 1394 | 4 | Slash pine | 252 |
| 5 | Grass-pasture | 483 | 5 | Painted metal sheets | 1345 | 5 | Fallow_smooth | 2678 | 5 | Oak/Broadleaf | 161 |
| 6 | Grass-trees | 730 | 6 | Bare Soil | 5029 | 6 | Stubble | 3959 | 6 | Hardwood | 229 |
| 7 | Grass-pasture-mowed | 28 | 7 | Bitumen | 1330 | 7 | Celery | 3579 | 7 | Swamp | 105 |
| 8 | Hay-windrowed | 478 | 8 | Self-Blocking Bricks | 3682 | 8 | Grapes_untrained | 11,271 | 8 | Graminiod marsh | 431 |
| 9 | Oats | 20 | 9 | Shadows | 947 | 9 | Soil_vinyard_develop | 6203 | 9 | Spartina marsh | 520 |
| 10 | Soybean-notill | 972 | 10 | Background | 164,624 | 10 | Corn_senesced_green_weeds | 3278 | 10 | Catiail marsh | 404 |
| 11 | Soybean-mintill | 2455 | | | | 11 | Lettuce_romaine_4wk | 1068 | 11 | Salt marsh | 419 |
| 12 | Soybean-clean | 593 | | | | 12 | Lettuce_romaine_5wk | 1927 | 12 | Mud flats | 503 |
| 13 | Wheat | 205 | | | | 13 | Lettuce_romaine_6wk | 916 | 13 | Water | 927 |
| 14 | Woods | 1265 | | | | 14 | Lettuce_romaine_7wk | 1070 | 14 | Background | 56,975 |
| 15 | Buildings-Grass-Trees-Drives | 386 | | | | 15 | Vinyard_untrained | 7268 | | | |
| 16 | Stone-Steel-Towers | 93 | | | | 16 | Vinyard_vertical_trellis | 1807 | | | |
| 17 | Background | 10,776 | | | | 17 | Background | 56,975 | | | |
| | Total Samples | 21,025 | | Total Samples | 207,400 | | Total Samples | 111,104 | | Total Samples | 314,368 |

**Indian Pines (IP)** is taken by the AVIRIS sensors. The size of this HSI dataset is 145 × 145, including 200 spectral bands and 24 noisy bands that cannot be reflected by water. It has 10,249 pixels that can be divided into 16 land categories for HSI classification. The False Color image and Ground Truth map are shown in Figure 4.

**Pavia University (PU)** is taken by the ROSIS sensors. The size of this image data is 615 × 345, which includes 9 categories. This PU dataset only leaves 103 bands and removes 12 bands. The False Color image and Ground Truth map are shown in Figure 5.

**Salinas (SA)** is taken by the AVIRIS sensors. It remains 204 bands after removing the noisy bands in HSI. The dataset size of SA is 512 × 217, which holds 54,129 pixels that can be divided into 16 land categories with true labels. The False Color image and Ground Truth map are shown in Figure 6.

**Kennedy Space Center (KSC)** is taken by the AVIRIS sensors. The size of KSC data is 512 × 614, which contains 13 land categories with true labels in total. This dataset only leaves 176 bands and removes 48 bands that cannot be reflected by water. The False Color image and Ground Truth map are shown in Figure 7.

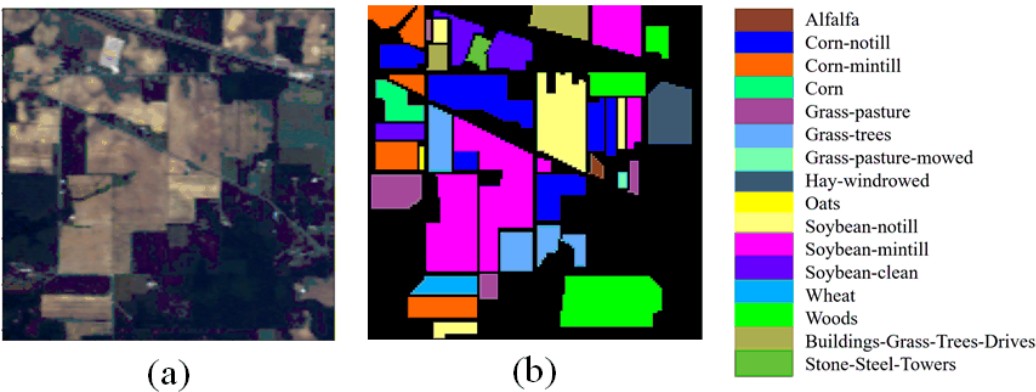

**Figure 4.** Indian Pines (IP) dataset. (**a**) False Color image. (**b**) Ground Truth map.

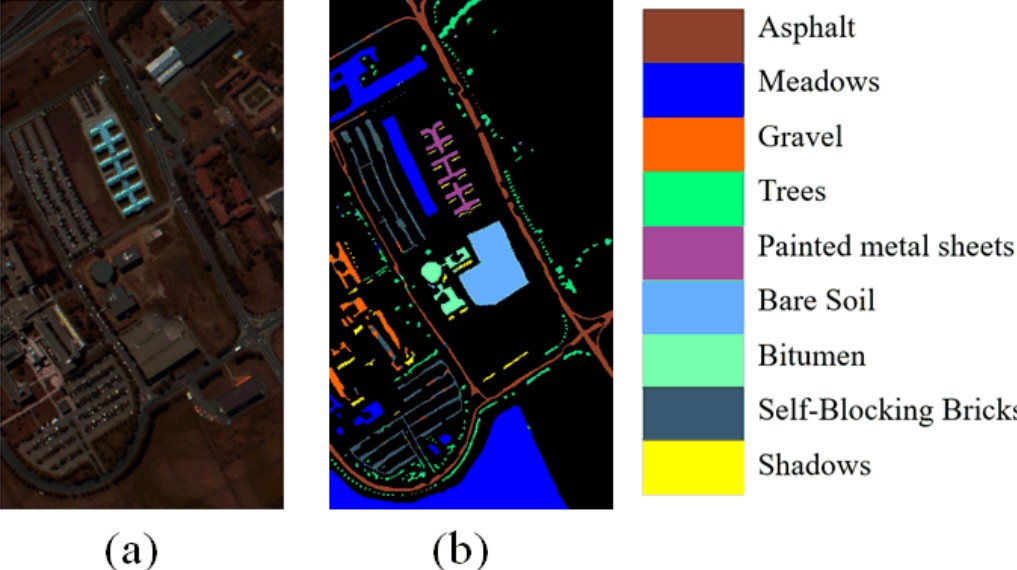

**Figure 5.** Pavia University (PU) dataset. (**a**) False Color image. (**b**) Ground Truth map.

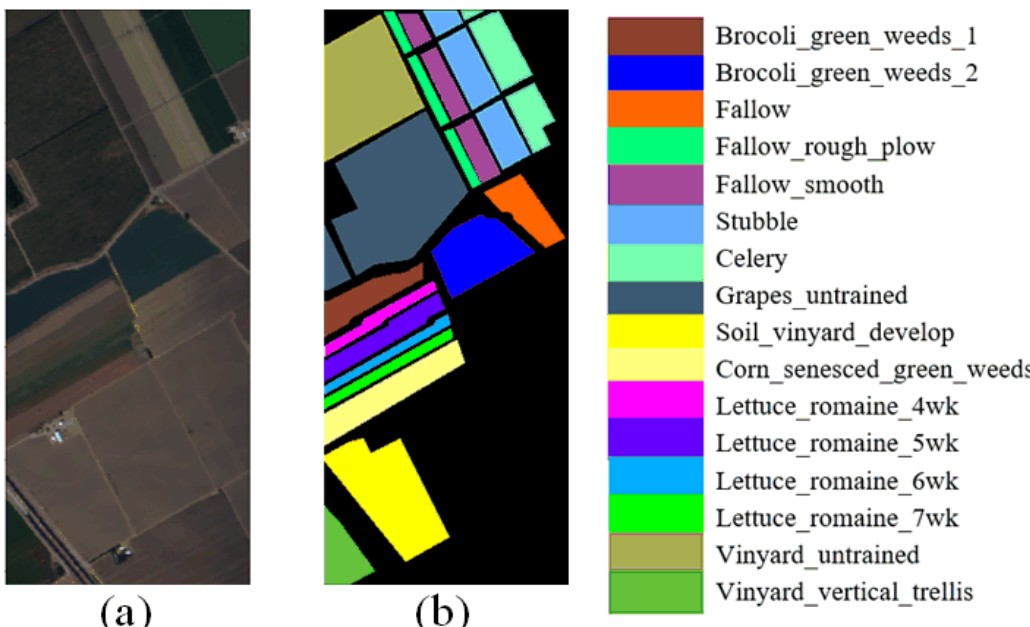

**Figure 6.** Salinas (SA) dataset. (**a**) False Color image. (**b**) Ground Truth map.

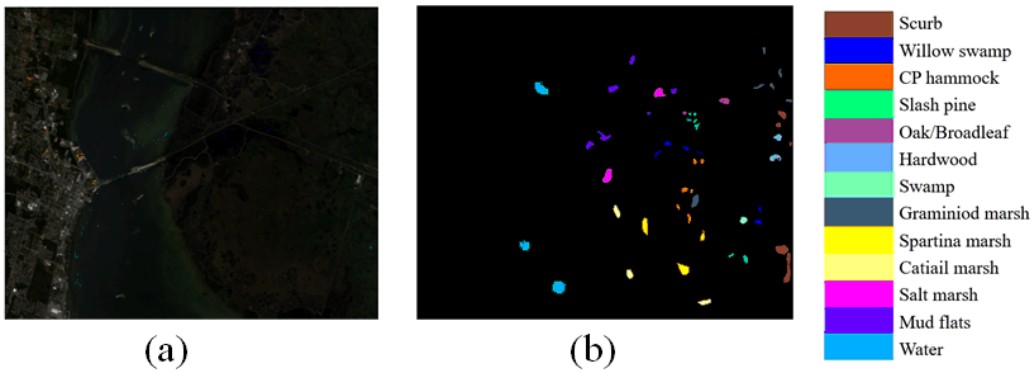

**Figure 7.** Kennedy Space Center (KSC) dataset. (**a**) False Color image. (**b**) Ground Truth map.

In addition to reducing the spectral bands of the PU by PCA to 90, the spectral bands of the KSC by PCA to 120, and the spectral bands of the IP and SA dataset by PCA to 110, we also set some other parameters. During the training process, we set the learning rate, training epochs, and patch size to 0.001, 200, and 13, respectively. Note that due to the reconstruction of the HSI dataset during training, we still chose the usual training ratio in these four datasets, which is set to 0.05, 0.1, 0.005, and 0.2, respectively. In the reconstructed datasets experiments, we additionally select some training samples from the original dataset to compute the knowledge loss that is used in the SKDN training. In the original datasets experiments, we additionally choose some training samples from the reconstructed dataset to compute the knowledge loss. In addition, three well-known numerical indicators, namely overall accuracy (OA), average accuracy (AA), and kappa coefficient (kappa), are used to evaluate the performance of different classification methods. OA is the ratio of the number of correctly classified samples to the total number of test samples, AA is the average accuracy across the accuracy of all classes, and Kappa is an available measure of agreement between ground truth and classification maps. All experiments were performed using four patch-data-based CNN methods in HSI classification, which are DPRN [10], CEGCN [1], SSTFF [23] and MRViT [26]. All results of these experiments in four real-world datasets are shown in Tables 2–11 and Figures 8–11.

**Table 2.** Classification results' OA from the improved methods and the original methods on four reconstructed datasets.

| Dataset | Random Ratio | Training Ratio | DPRN | SKDN(DPRN) | SSTFF | SKDN(SSTFF) | MRViT | SKDN(MRViT) |
|---------|-------------|----------------|------|-----------|-------|------------|-------|------------|
| IP | 0% | 5% | 95.17 | **95.32** | 96.39 | **96.46** | 98.08 | **98.12** |
|  | 20% | 5% | 65.4 | **76.85** | 68.94 | **74.05** | 73.72 | **79.22** |
|  | 50% | 5% | 32.61 | **45.49** | 37.61 | **40.46** | 43.34 | **50.43** |
| PU | 0% | 10% | 95.02 | **99.91** | 99.12 | **99.77** | 99.08 | **99.51** |
|  | 20% | 10% | 81.88 | **87.25** | 80.23 | **80.78** | 81.6 | **85.19** |
|  | 50% | 10% | 64.51 | **71.87** | 53.59 | **54.78** | 62.11 | **66.79** |
| SA | 0% | 0.50% | 95.58 | **95.92** | 96.39 | **96.9** | 96.27 | **96.73** |
|  | 20% | 0.50% | 55.73 | **73.63** | 70.89 | **76.38** | 74.5 | **78.51** |
|  | 50% | 0.50% | 24.96 | **36.66** | 32.57 | **45.48** | 40.4 | **51.12** |
| KSC | 0% | 20% | 97.01 | **97.82** | 98.08 | **98.1** | 98.17 | **98.21** |
|  | 20% | 20% | 57.73 | **83.41** | 70.37 | **76.17** | 81.75 | **82.38** |
|  | 50% | 20% | 26.69 | **55.36** | 43.05 | **47.5** | 50.26 | **50.91** |

**Table 3.** Classification results' AA from the improved methods and the original methods on four reconstructed datasets.

| Dataset | Random Ratio | Training Ratio | DPRN | SKDN(DPRN) | SSTFF | SKDN(SSTFF) | MRViT | SKDN(MRViT) |
|---------|-------------|----------------|------|-----------|-------|------------|-------|------------|
| IP | 0% | 5% | 92.00 | **93.01** | 92.68 | **93.56** | 93.39 | **95.26** |
|  | 20% | 5% | 54.49 | **67.26** | 61.98 | **70.21** | 64.55 | **74.24** |
|  | 50% | 5% | 17.63 | **31.85** | 30.41 | **31.98** | 32.11 | **44.69** |
| PU | 0% | 10% | 90.3 | **95.66** | 98.44 | **98.72** | 98.65 | **99.03** |
|  | 20% | 10% | 73.21 | **84.16** | 76.22 | **76.84** | 77.02 | **84.52** |
|  | 50% | 10% | 47.09 | **61.27** | 41.8 | **45.83** | 47.78 | **60.17** |
| SA | 0% | 0.50% | 95.32 | **95.44** | 96.12 | **96.63** | 96.09 | **96.72** |
|  | 20% | 0.50% | 54.21 | **74.12** | 71.64 | **74.65** | 72.92 | **75.91** |
|  | 50% | 0.50% | 23.24 | **33.81** | 28.74 | **42.07** | 37.27 | **48.36** |
| KSC | 0% | 20% | 96.5 | **97.23** | 97.65 | **98.03** | 97.37 | **97.85** |
|  | 20% | 20% | 48.61 | **81.76** | 69.3 | **69.65** | 79.43 | **80.05** |
|  | 50% | 20% | 18.28 | **51.7** | 38.36 | **43.93** | 45.43 | **50.37** |

**Table 4.** Classification results' kappa from the improved methods and the original methods on four reconstructed datasets.

| Dataset | Random Ratio | Training Ratio | DPRN | SKDN(DPRN) | SSTFF | SKDN(SSTFF) | MRViT | SKDN(MRViT) |
|---------|-------------|----------------|------|-----------|-------|------------|-------|------------|
| IP | 0% | 5% | 95.82 | **95.91** | 95.88 | **96.22** | 97.37 | **97.81** |
|  | 20% | 5% | 60.44 | **73.60** | 64.44 | **70.41** | 69.42 | **74.81** |
|  | 50% | 5% | 21.72 | **36.87** | 28.60 | **30.01** | 34.88 | **43.01** |
| PU | 0% | 10% | 93.47 | **95.61** | 99.53 | **99.60** | 98.42 | **98.91** |
|  | 20% | 10% | 75.34 | **83.05** | 73.77 | **74.51** | 75.51 | **77.87** |
|  | 50% | 10% | 50.22 | **61.93** | 36.76 | **38.54** | 48.53 | **58.02** |
| SA | 0% | 0.50% | 95.31 | **95.42** | 96.23 | **96.43** | 96.36 | **95.82** |
|  | 20% | 0.50% | 50.34 | **70.75** | 67.80 | **73.70** | 71.51 | **75.45** |
|  | 50% | 0.50% | 17.47 | **29.97** | 24.37 | **38.68** | 32.51 | **43.92** |
| KSC | 0% | 20% | 96.57 | **97.01** | 97.53 | **97.97** | 97.62 | **98.27** |
|  | 20% | 20% | 53.11 | **81.47** | 67.02 | **67.83** | 74.66 | **78.27** |
|  | 50% | 20% | 20.39 | **50.18** | 35.92 | **41.46** | 44.45 | **46.29** |

**Table 5.** The experimental results' OA from the improved models and the original models on four real-world datasets.

| Dataset | Random Ratio | Training Ratio | DPRN | SKDN(DPRN) | SSTFF | SKDN(SSTFF) | MRViT | SKDN(MRViT) |
|---------|-------------|----------------|------|-----------|-------|------------|-------|------------|
| IP | 0% | 5% | 95.17 | **95.32** | 96.39 | **96.46** | 98.08 | **98.12** |
|  | 20% | 5% | 92.61 | **95.04** | 83.3 | **94.65** | 91.74 | **97.24** |
|  | 50% | 5% | 74.45 | **82.21** | 52.49 | **94.09** | 76.76 | **96.48** |
| PU | 0% | 10% | 95.02 | **99.91** | 99.12 | **99.77** | 99.08 | **99.51** |
|  | 20% | 10% | 95.27 | **99.42** | 95.51 | **99.61** | 96.16 | **98.98** |
|  | 50% | 10% | 92.25 | **96.37** | 85.66 | **98.61** | 92.37 | **97.15** |
| SA | 0% | 0.50% | 95.58 | **95.92** | 96.39 | **96.9** | 96.27 | **96.73** |
|  | 20% | 0.50% | 90.12 | **92.89** | 89.66 | **96.01** | 91.12 | **96.24** |
|  | 50% | 0.50% | 84.07 | **89.01** | 60.33 | **95.93** | 77.15 | **94.07** |
| KSC | 0% | 20% | 97.01 | **97.82** | 98.08 | **98.1** | 98.17 | **98.21** |
|  | 20% | 20% | 95.13 | **99.44** | 34.71 | **65.79** | 91.49 | **94.23** |
|  | 50% | 20% | 71.28 | **77.14** | 17.66 | **57.25** | 77.01 | **85.61** |

**Table 6.** The experimental results' AA from the improved models and the original models on four real-world datasets.

| Dataset | Random Ratio | Training Ratio | DPRN | SKDN(DPRN) | SSTFF | SKDN(SSTFF) | MRViT | SKDN(MRViT) |
|---------|--------------|----------------|------|------------|-------|-------------|-------|-------------|
| IP | 0% | 5% | 92.81 | **93.04** | 92.86 | **93.73** | 96.48 | **97.66** |
| | 20% | 5% | 88.62 | **89.15** | 84.19 | **88.39** | 92.64 | **94.97** |
| | 50% | 5% | 69.44 | **76.84** | 83.65 | **84.79** | 80.73 | **85.89** |
| PU | 0% | 10% | 98.79 | **99.53** | 99.06 | **99.24** | 98.67 | **99.44** |
| | 20% | 10% | 98.26 | **99.31** | 94.62 | **99.09** | 96.1 | **98.59** |
| | 50% | 10% | 94.62 | **95.04** | 87.44 | **97.85** | 93.34 | **96.18** |
| SA | 0% | 0.50% | 92.12 | **94.79** | 95.69 | **96.55** | 95.51 | **96.79** |
| | 20% | 0.50% | 90.93 | **92.84** | 89.33 | **96.39** | 90.75 | **97.21** |
| | 50% | 0.50% | 83.71 | **88.79** | 60.12 | **94.51** | 72.46 | **95.68** |
| KSC | 0% | 20% | 95.01 | **96.57** | 92.18 | **93.59** | 92.04 | **92.96** |
| | 20% | 20% | 93.22 | **96.16** | 57.34 | **59.16** | 90.76 | **90.91** |
| | 50% | 20% | 73.78 | **76.3** | 41.45 | **45.26** | 90.18 | **91.45** |

**Table 7.** The experimental results' kappa from the improved models and the original models on four real-world datasets.

| Dataset | Random Ratio | Training Ratio | DPRN | SKDN(DPRN) | SSTFF | SKDN(SSTFF) | MRViT | SKDN(MRViT) |
|---------|--------------|----------------|------|------------|-------|-------------|-------|-------------|
| IP | 0% | 5% | 93.68 | **94.10** | 94.09 | **94.71** | 96.33 | **97.21** |
| | 20% | 5% | 93.17 | **94.35** | 93.19 | **94.65** | 94.44 | **96.85** |
| | 50% | 5% | 74.29 | **79.96** | 92.23 | **93.26** | 93.89 | **95.97** |
| PU | 0% | 10% | 96.73 | **98.77** | 97.01 | **99.53** | 97.74 | **99.03** |
| | 20% | 10% | 97.16 | **98.23** | 96.78 | **99.48** | 96.83 | **98.65** |
| | 50% | 10% | 94.74 | **95.15** | 89.95 | **98.15** | 90.41 | **96.21** |
| SA | 0% | 0.50% | 93.27 | **95.11** | 94.22 | **96.24** | 94.33 | **96.08** |
| | 20% | 0.50% | 92.5 | **92.09** | 93.18 | **95.56** | 92.09 | **95.81** |
| | 50% | 0.50% | 85.93 | **87.77** | 70.57 | **95.01** | 90.93 | **93.41** |
| KSC | 0% | 20% | 96.07 | **97.35** | 85.66 | **89.73** | 95.76 | **97.73** |
| | 20% | 20% | 84.51 | **97.24** | 60.14 | **61.55** | 89.18 | **93.57** |
| | 50% | 20% | 70.37 | **74.41** | 50.49 | **51.57** | 83.62 | **87.94** |

**Table 8.** The experimental results from different single-pixel-based models on Indian Pines.

| Methods | | DPRN | CEGCN | SSTFF | MRViT | SKDN(MRViT) |
|---------|---|------|-------|-------|-------|-------------|
| Random Shuffle | Training Ratio | OA | OA | OA | OA | OA |
| 0% | 5% | 68.84 | **97.61** | 62.55 | 65.19 | 74.51 |
| 20% | 5% | 64.49 | **73.81** | 63.61 | 67.86 | 72.86 |
| 50% | 5% | 67.80 | 34.24 | 64.94 | 66.36 | **74.83** |
| 100% | 5% | 68.75 | 19.24 | 64.93 | 65.37 | **73.17** |
| 20% | 20% | **87.74** | 75.6 | 80.87 | 83.62 | 86.31 |
| 50% | 20% | **87.43** | 47.47 | 79.96 | 83.52 | 85.79 |
| 100% | 20% | **87.14** | 19.45 | 79.84 | 84.68 | 86.47 |
| 20% | 50% | 90.55 | 74.56 | 90.68 | 93.14 | **95.97** |
| 50% | 50% | 90.67 | 48.4 | 90.35 | 92.45 | **94.75** |
| 100% | 50% | 90.18 | 24.31 | 87.01 | 92.31 | **94.39** |

**Table 9.** The experimental results from different single-pixel-based models on Pavia University.

| Methods | | DPRN | CEGCN | SSTFF | MRViT | SKDN(MRViT) |
|---------|---|------|-------|-------|-------|-------------|
| Random Shuffle | Training Ratio | OA | OA | OA | OA | OA |
| 0% | 10% | 94.12 | **96.73** | 90.66 | 84.06 | 84.79 |
| 20% | 10% | **93.82** | 83.5 | 91.82 | 84.76 | 88.77 |
| 50% | 10% | 92.01 | 64.29 | **92.22** | 85.62 | 88.14 |
| 100% | 10% | **93.2** | 50.89 | 91.39 | 85.97 | 87.39 |
| 20% | 30% | 95.0 | 83.78 | **95.71** | 92.1 | 93.66 |
| 50% | 30% | 94.72 | 65.16 | **95.37** | 92.34 | 93.17 |
| 100% | 30% | 94.5 | 55.33 | **95.05** | 92.06 | 94.09 |
| 20% | 50% | 95.85 | 81.98 | 96.57 | 96.28 | **96.75** |
| 50% | 50% | 94.38 | 60.95 | **96.9** | 96.31 | 95.73 |
| 100% | 50% | 94.16 | 56.45 | 96.64 | 96.18 | **96.71** |

**Table 10.** The experimental results from different single-pixel-based models on Salinas.

| Methods | | DPRN | CEGCN | SSTFF | MRViT | SKDN(MRViT) |
|---|---|---|---|---|---|---|
| Random Shuffle | Training Ratio | OA | OA | OA | OA | OA |
| 0% | 0.50% | 84.77 | **99.13** | 85.84 | 76.31 | 77.87 |
| 20% | 0.50% | 85.93 | 80.13 | **86.51** | 78.76 | 82.79 |
| 50% | 0.50% | **84.86** | 53.05 | 84.52 | 75.58 | 83.52 |
| 100% | 0.50% | 84.07 | 18.98 | **84.56** | 76.16 | 82.71 |
| 50% | 5% | 88.95 | 78.58 | 92.76 | 90.47 | **92.77** |
| 20% | 5% | 89.12 | 87.72 | 92.74 | 90.99 | **93.18** |
| 100% | 5% | 88.44 | 64.71 | **92.81** | 90.86 | 92.27 |
| 20% | 20% | 93.73 | 91.94 | **95.74** | 94.29 | 95.57 |
| 50% | 20% | **95.12** | 89.96 | 95.03 | 94.14 | 95.01 |
| 100% | 20% | 94.79 | 83.12 | **95.56** | 94.15 | 94.35 |
| 20% | 50% | 95.16 | 92.55 | 97.14 | 97.23 | **97.59** |
| 50% | 50% | 94.71 | 90.65 | 97.59 | 97.45 | **97.74** |
| 100% | 50% | 94.84 | 88.65 | **97.69** | 97.22 | 97.48 |

**Table 11.** The experimental results from different single-pixel-based models on Kennedy Space Center.

| Methods | | DPRN | CEGCN | SSTFF | MRViT | SKDN(MRViT) |
|---|---|---|---|---|---|---|
| Random Shuffle | Training Ratio | OA | OA | OA | OA | OA |
| 0% | 20% | 93.13 | **95.86** | 32.71 | 88.76 | 89.59 |
| 20% | 20% | **90.52** | 77.88 | 11.77 | 79.25 | 81.88 |
| 50% | 20% | **85.96** | 52.99 | 15.5 | 75.88 | 80.41 |
| 100% | 20% | **87.79** | 13.93 | 16.68 | 80.2 | 79.74 |
| 20% | 40% | **91.43** | 78.86 | 15.36 | 87.4 | 90.82 |
| 50% | 40% | 90.29 | 54.74 | 15.2 | 82.22 | **90.38** |
| 100% | 40% | 90.14 | 18.24 | 8.75 | 84.57 | **90.43** |
| 20% | 60% | 93.66 | 78.65 | 8.64 | 93.98 | **94.01** |
| 50% | 60% | 92.88 | 56.89 | 14.89 | 93.55 | **93.71** |
| 100% | 60% | 92.17 | 20.94 | 15.23 | 92.82 | **93.52** |

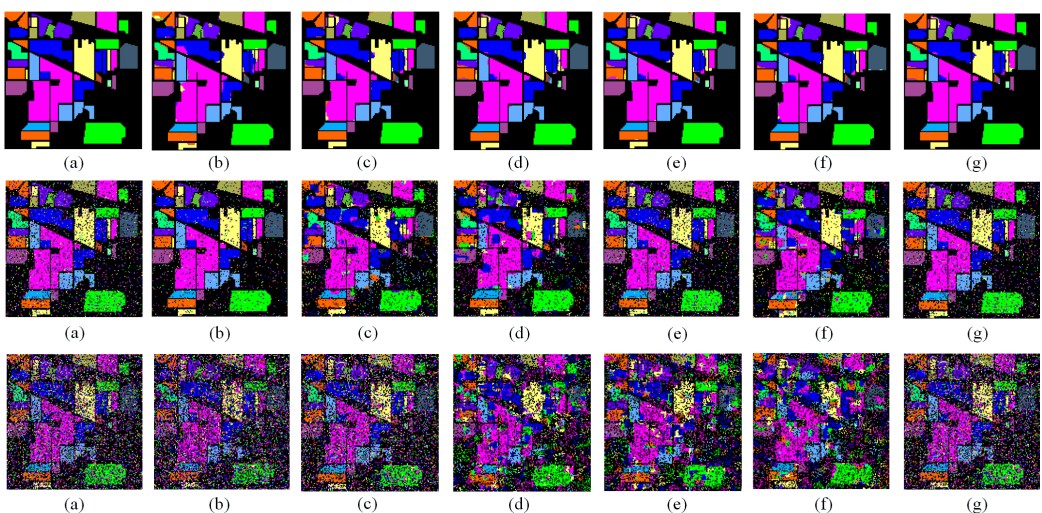

**Figure 8.** The first row is the prediction map of Indian Pines (IP) with a random shuffling ratio of 0, the second row is the experimental results of IP with a random shuffling proportion of 0.2, and the third row is the experimental results of IP with a random shuffling ratio of 0.5. (**a**) Ground Truth map; (**b**) DPRN; (**c**) SKDN(DPRN); (**d**) SSFTT; (**e**) SKDN(SSTFF); (**f**) MRViT; (**g**) SKDN(MRViT).

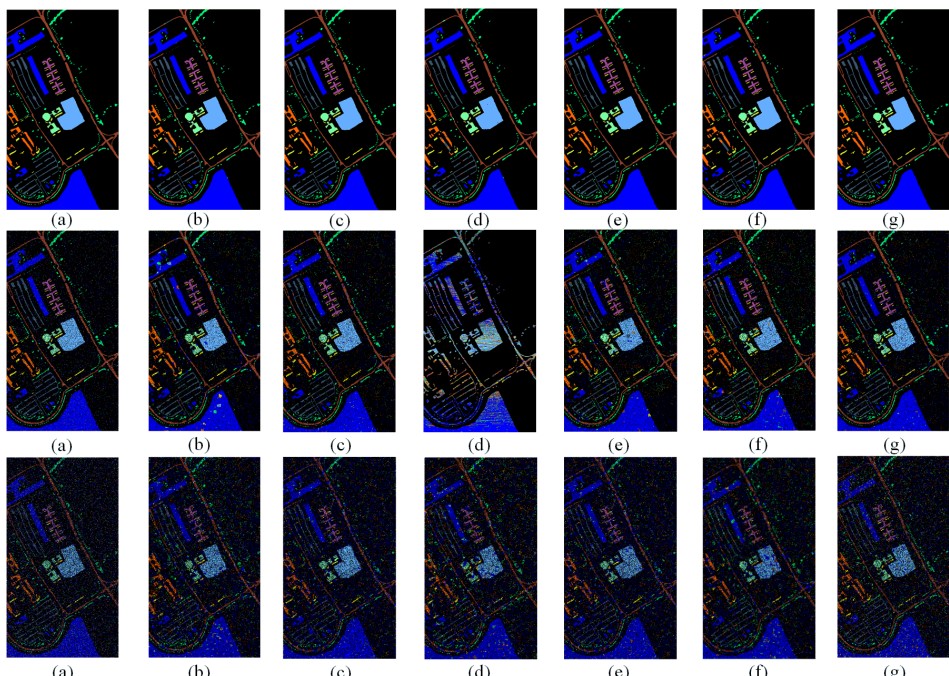

**Figure 9.** The first row is the prediction map of Pavia University (PU) with a random shuffling ratio of 0, the second row is the experimental results of PU with a random shuffling proportion of 0.2, and the third row is the experimental results of PU with a random shuffling ratio of 0.5. (**a**) Ground Truth map; (**b**) DPRN; (**c**) SKDN(DPRN); (**d**) SSFTT; (**e**) SKDN(SSTFF); (**f**) MRViT; (**g**) SKDN(MRViT).

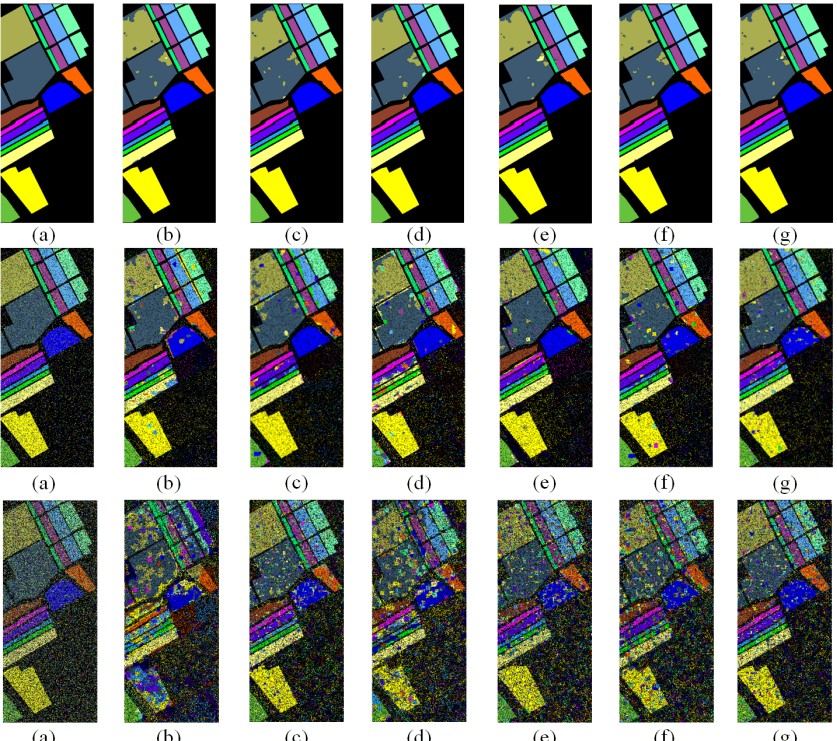

**Figure 10.** The first row is the prediction map of Salinas (SA) with a random shuffling ratio of 0, the second row is the experimental results of SA with a random shuffling proportion of 0.2, and the third row is the experimental results of SA with a random shuffling ratio of 0.5. (**a**) Ground Truth map; (**b**) DPRN; (**c**) SKDN(DPRN); (**d**) SSFTT; (**e**) SKDN(SSTFF); (**f**) MRViT; (**g**) SKDN(MRViT).

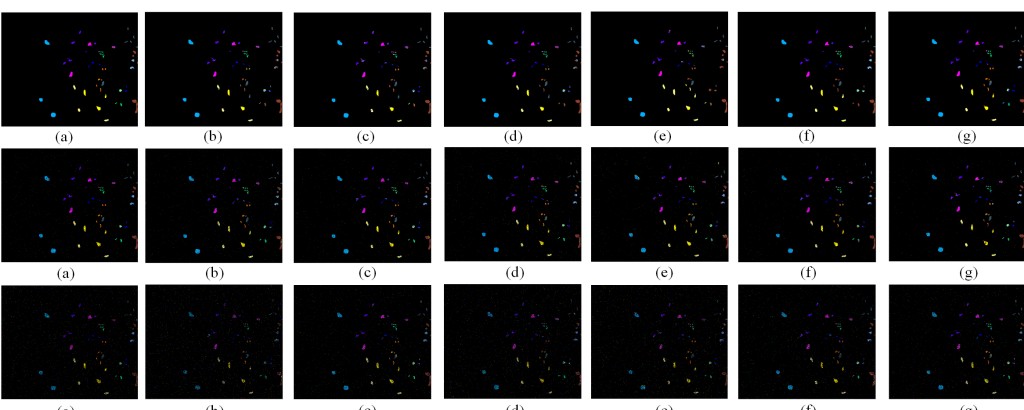

**Figure 11.** The first row is the prediction map of KSC with a random shuffling ratio of 0, the second row is the experimental results of KSC with a random shuffling proportion of 0.2, and the third row is the experimental results of KSC with a random shuffling ratio of 0.5. (**a**) Ground Truth map; (**b**) DPRN; (**c**) SKDN(DPRN); (**d**) SSFTT; (**e**) SKDN(SSTFF); (**f**) MRViT; (**g**) SKDN(MRViT).

*3.1. The Experimental Results of the SKDN on Reconstructed Data Based Four Real-World Datasets*

Since our random shuffling strategy disrupts the data homogeneity of the original patch data, we introduce the designed sub-branch into all original networks to form it improved network SKDN, and we complete these experiments on four reconstructed datasets. Note that we additionally chose some training samples from the original datasets for the improved model (SKDN)'s training, where the proportions of these original datasets in the model's training are the same as the ratio in reconstructed datasets. In these experiments, we set the initial values of $\alpha$ and $\beta$ to 0.5 in the weighted cross-entropy loss function when the random shuffling ratio was 0%. Tables 2–4 report the experimental results of the networks on these four datasets, and the visual results of these networks can be seen in Figure 8 through Figure 11.

(1) **The analysis of the experiments on the reconstructed dataset based on Indian Pines:** the second row in Table 2 lists the quantitative results of the different methods, and Figure 8 shows the prediction maps corresponding to these methods for the reconstructed IP dataset. Compared to the network DPRN, the SKDN(DPRN) model improves the OA by 11.45% and achieves a better visual result (Figure 8c) when the random ratio is 20%. It is 12.88% higher than DPRN when the random mixture ratio is 50%. This shows that the result of the improved model is better than that of the original model, which proves the effectiveness of RFL in the network. Obviously, SKDN(SSTFF) also has a better classification performance compared to SSTFF. The value of OA is 5.11% higher than that of SSTFF when the random shuffling ratio is 20%. It is 2.85% higher than that of SSTFF when the ratio of the random mixture is 50%. From Figure 8f, it is easy to see that the improved model can extract more discriminative local context information and category information in the patch data. The corresponding values of the SKDN(MRViT) model in Table 2 also show that the improved network performs better than MRViT. The value of OA is 5.5% higher than MRViT when the random mixture proportion is 20%. It is 7.09% higher than MRViT when the proportion of the random mixture is 50%. This also proves that the designed sub-branch RFL is more effective in the reconstructed dataset.

(2) **The analysis of the experiments on the reconstructed Pavia University dataset:** the third row in Table 2 illustrates the quantitative results of the different methods, and Figure 8 shows the prediction maps corresponding to these methods on for reconstructed PU dataset. It is clear that our improved-method SKDN(DPRN) performs better than the original method DPRN. The value of the OA is 5.37% higher than that of DPRN when the random mixture proportion is 20%. It is 7.36% higher than that of DPRN when the proportion of random mixture is 50%. By analyzing these visual maps, it is easy to find that SKDN(DPRN) in Figure 9c has a lower prediction error range than DPRN in

Figure 9b. In contrast to SSTFF, the improved SKDN(SSTFF) model in Figure 9e also has fewer misclassifications. This can be confirmed by the improved OA value (0.55% and 1.19%) in Table 2. Figure 9g also shows that the improved SKDN(MRViT) method has better performance, which is also confirmed by its OA value (85.19% and 66.79%) in Table 2. This is direct evidence that the sub-branching used to construct the SKDN is effective.

**(3) The analysis of the experiments on the reconstructed Salinas dataset:** the fourth row in Table 2 shows the classification results of the different methods. The corresponding prediction graphs for the reconstructed SA dataset are shown in Figure 10. Compared with the first network, the improved-network SKDN(DPRN) improves the OA by 17.9% when the random shuffling ratio is 20% and achieves a better classification result with fewer misclassified regions in the second row in Figure 10c. It is 11.7% higher than DPRN when the proportion of random shuffling is 50%. These results show that RFL is useful for the model. It is clear that the experimental results of SKDN(SSTFF) are superior to those of SSTFF, which can also be seen in Figure 10e and is confirmed by the better OA value (5.49% and 12.91%) in Table 2. In Figure 10f,g, the experimental results of SKDN(MRViT) are superior to those of MRViT with fewer training samples. The SKDN(MRViT) model improves OA by 4.01% when the random ratio is 20%. It is 10.72% higher than MRViT when the proportion of random shuffling is 50%. These results can prove that RFL can extract more discriminative local context information in the new dataset.

**(4) The analysis of the experiments on the reconstructed dataset Kennedy Space Center:** the fifth row in Table 2 shows the quantitative results of different methods, and Figure 11 is the prediction maps corresponding to these methods on the reconstructed KSC dataset. It is noticeable that the OA value in the experimental results of the improved model is better than that of the original model. In the following, the results of these methods are only briefly analyzed. Compared with the network DPRN, the model SKDN(DPRN) improves OA by 25.68% and achieves a better result, as shown in Figure 11c, where the proportion of random shuffling is 20%. It is 28.67% higher than the DPRN when the proportion of random mixture is 50%. Moreover, the classification capability of SKDN(SSTFF) is better than that of SSTFF, which is also evidenced by the better OA value (5.8% and 4.45%). Compared to the MRViT, SKDN(MRViT) also achieves better classification performance (82.38% and 50.91%) in Table 2. SKDN(MRViT) has less misclassification than MRViT in its domain, which can be seen in Figure 11g. All the results of these experiments proved that our model SKDN can indeed extract more local context information and category information from random shuffling patch data for HSI classification. Our proposed sub-branch RFL, which fuses the loss rates of two sub-branches in its loss function, can effectively improve the classification accuracy in random shuffling data. All results of these experiments have proven that our model SKDN can really extract more local context information and category information from random shuffling patch data for HSI classification. Our proposed sub-branch RFL, which fuses the loss rates of two sub-branches in its loss function, can effectively improve the classification accuracy in random shuffling data.

*3.2. The Experimental Results of the SKDN on Four Real-World Datasets*

To further validate the usefulness of the proposed sub-branch RFL, we introduce RFL into the original network to form its improved network SKDN and let the improved model and the original model achieve the HSI classification task on the four original datasets. Note that we additionally chose some training samples from the reconstructed datasets for the improved model (SKDN) training, where the proportions of these reconstructed datasets in model training are the same as the ratio in the original datasets. Table 5 lists the quantitative results of the original methods and the improved methods on four original datasets. We also set the initial values of $\alpha$ and $\beta$ to 0.5 in the weighted cross-entropy loss function when the random shuffling ratio was 0%. Hereinafter, we just briefly analyze the results of these methods on four original datasets.

The second row in Table 5 lists the experimental results of the original networks and the improved models for the IP dataset. The classification result of SKDN(DPRN) is

2.43% higher than the result of DPRN for OA when the proportion of random mixture is 20%; the value of the OA also improves by 7.76% when the random shuffling ratio is 50%. Compared with the network SSTFF, the improved model SKDN(SSTFF) has a better classification performance, which can be verified by its improved OA value (11.35% and 41.6%) in Table 5. We also find that the improved network SKDN(MRViT) (97.24% and 96.48%) performs better than MRViT. This proves that the proposed RFL is effective in the original dataset.

The third row in Table 5 shows the experimental results of the original networks and the improved models for the PU dataset. Compared with the original method DPRN, the improved-method SKDN(DPRN) improves OA by 4.15% when the random shuffling ratio is 20%. It is 4.12% higher than DPRN when the proportion of random shuffling is 50%. We also can see that the improved-method SKDN(SSTFF) also performs better than SSTFF, which can also be verified by its OA value (99.61% and 98.61%). Compared to the MRViT, SKDN(MRViT) also achieves a better classification performance, which can be verified by its improved OA value (2.82% and 4.78%). These results demonstrate that the RFL is useful in our model.

The fourth row in Table 5 shows the experimental results of the original networks and the improved networks for the SA dataset. Compared with DPRN, the improved method SKDN(DPRN) improves OA by 2.77% when the random shuffling ratio is 20%. It is 4.94% higher than DPRN when the proportion of random shuffling is 50%. It is shown that the improved-method SKDN(SSTFF) also performs better than SSTFF, which can be verified by its improved OA value (6.35% and 35.6%). Compared to the MRViT, SKDN(MRViT) also achieves a better classification performance, which can also be seen in its OA value (96.24% and 94.07%) in Table 5. These results also prove that the RFL can capture the discriminative feature in the original dataset.

The fifth row in Table 5 lists the experimental results of the original networks and the improved networks for the KSC dataset. Compared with DPRN, the OA of SKDN(DPRN) improves by 4.31% and 5.86% when the random shuffling ratios are 20% and 50%, respectively. Moreover, the OA of SKDN(SSTFF) is 31.08% higher than that of SSTFF when the proportion of random shuffling is 20%. It is 39.59% higher than SSTFF when the random shuffling ratio is 50%. Compared to the MRViT, FSKDN(MRViT) also achieves a better classification performance, which can be verified by its improved OA value (2.74% and 8.6%). All results of these experiments show that the classification capacity of the improved model SKDN is superior to that of the original model, which in turn proves that the proposed RFL sub-branch is effective and feasible.

*3.3. The Experimental Results of Single Pixel-Based Methods on Reconstructed Data Based on Four Real-World Datasets*

We set the value of the patch-size parameter to 1 in the patch-data-based CNN methods in the HSI classification so that they become the single-pixel-based methods in this experiment. Tables 8–11 show the results of each network with different random shuffling ratios and training ratios in the HSI classification. We see that the accuracy of these models improves slightly when the training fraction is increased and the random mixture ratio is kept constant. However, when the same training fraction is kept and the random mixture ratio is increased, the classification results of the CNN-based method and the Transformer-based method, such as the DPRN and the SSTFF, vary only within a certain value interval, and the OA value does not change significantly, but the classification results of the GCN-based network show a downward trend. Using the experiments of the reconstructed dataset based on IP as an analysis case and setting the proportion of each method to 5% in the model's training, we find that the results of the DPRN are in the range of 64% to 69% when the proportion of random shuffling increases, the results of the SSTFF are in the range of 62% to 65%, the results of the MRViT are in the range of 65% to 68%, and the results of the SKDN(MRViT) are in the range of 72% to 75%. However, the results of the CEGCN decreased from 97% to 19%. These experiments show that no matter how the

proportion of random shuffling increases, the classification accuracy results of the single pixel-based models for HSI classification do not change significantly. This may indicate that the single pixel-based methods do not use the local context relationship between pixels for the classification of HSIs, which also proves that the strategy of reconstructing the dataset has no major effect on the experimental classification results of the single pixel-based networks for the classification of HSIs.

## 4. Discussion

Several conclusions can be drawn regarding the role of the random shuffle strategy and the sub-branch RFL. First, the classification results of these experiments on the reconstructed datasets and original datasets prove that our model SKDN can extract more local context information and category information from random shuffling patch data for HSI classification. Furthermore, these experiments reveal that the patch-data-based CNN methods really depend on the data homogeneity of patch-data in the training process. This can also be verified by the classification results of these original models that present a downward trend as the proportion of random shuffling increases. What is more, the RFL is introduced into the original model to construct the SKDN, and the new loss function in the SKDN can fuse loss values from two sub-branches that enhance the recognition ability of the original method. Although the combination of the random shuffling strategy and the RFL cannot achieve a significant effect in the single pixel-based method's training process, it also enriches the related research on the single pixel-based methods in HSI classification. The work in this paper provides certain reference values for subsequent research into the patch-data-based CNN methods.

## 5. Conclusions

To resolve the issue of whether the patch-data-based CNN methods really depend on the data homogeneity of the patch-data in HSI, we propose a strategy of randomly shuffling pixel data to validate the influence of the characteristics of the patch-data in HSI classification networks. Specifically, it is to randomly assign the pixels in the original dataset to other locations. To ensure that the classification networks of HSI can learn more local contextual information and category information regarding the patch data, we also propose a sub-branch to fuse the loss rates. In detail, the loss rate calculated by this sub-branch in the new patch data is cross combined with the loss rate calculated by another branch in the original patch data to construct a new hyperspectral classification network named SKDN. Extensive experiments on these four datasets indicate that as the proportion of randomly shuffled data increases, the patch-data-based CNN methods are no longer able to extract more discriminative local context information for classifying pixels in HSI. However, the network SKDN constructed by our proposed sub-branch can effectively address this problem, which can improve the classification accuracy of randomly shuffled data. In addition, we also introduce the proposed sub-branch into the original network and let the improved model and the original model achieve HSI classification on the original dataset to explore the effectiveness of the sub-branch RFL. The results of all experiments show that the classification performance of the improved model is better than that of the original model, so they also prove that the proposed sub-branch is effective and feasible.

**Author Contributions:** Conceptualization, Z.Y. and Y.C.; methodology, Z.Y. and Y.C.; validation, Y.C.; formal analysis, Z.Y., Y.C., T.Z. and J.J.; writing—original draft preparation, Z.Y., Y.C., T.Z. and J.J.; funding acquisition, Z.Y., X.Z. and J.L. All authors have read and agreed to the published version of the manuscript.

**Funding:** This work was supported by the National Natural Science Foundation of China (No. 62261026, 62262026, 62003251, and 62201343), the Education Department Foundation of Jiangxi Province (No. GJJ2201357 and GJJ211111), the Jiangxi Natural Science Foundation (No. 20232ACB212006 and 20232BAB202020), the Key Laboratory of System Control and Information Processing, Ministry of Education (Scip202106), and the Shanghai Key Laboratory of Navigation and Location Based Services (No. SKLNLBS2023001).

**Data Availability Statement:** The data presented in this study are available on request from the corresponding author.

**Conflicts of Interest:** The authors declare no conflicts of interest. The founding sponsors had no role in the design of the study; in the collection, analyses, or interpretation of the data; in the writing of the manuscript; nor in the decision to publish the results.

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
