# Peer review of "Random Shuffling Data for Hyperspectral Image Classification with Siamese and Knowledge Distillation Network"

_remotesensing, doi:10.3390/rs15164078_

Round 1
Reviewer 1 Report
Summary of the paper:
The authors of this paper investigate the dependence of patch-based CNN methods on the local contextual information by introducing a "random shuffling strategy", and show that those CNNs perform worse on randomly shuffled, inhomogenious data.
Further, they propose a RFL sub-branch and construct a new hyperspectral classification network (SKDN) which shows improved classification accuracy on randomly shuffled data.
General comments:
Overall the paper is well-structured and nice to read. The English language is mostly fine. All relevant references are included.
However, there is a major conceptual weakness: The paper claims to address the dependence of patch-based CNNs on the homogenity of patch data by randomly shuffling the pixels in the data cube. But as the defining property of these kind of CNNs is indeed to rely on and make use of spatial local context information, how is it beneficial or even of interest to use randomly shuffeled data or to improve their classification accuracy on this kind of data?
Specific comments:
First, the title of the paper is based on that of the popular "SpectralFormer" paper, which is more pretentious than funny in my opinion.
In the Abstract, some things are unclear and might need to be explained better. For example,
in line 5, it is not fully clear what "homogenity" means in this context,
and in line 8, it does not become clear that "sub-branch" means branch in a siamese network structure.
Further, the abbreviations "RFL" and "SKDN" are not written out or explained at their first occurence, but they should.
The paper gives a nice and comprehensive introduction and overview on pixel-based and patch-based methods for hyperspectral image classification.
Only the part on GCNs is too verbose (proportionally) and contains many older references that are also not relevant for the remainder of the paper.
What in turn stands out positively is the paragraph on attention mechanisms and vision transformers, which is frequently missing in some earlier work.
The contributions part is somewhat fuzzy formulated.
In line 124 f., it says "the patch-data-based methods still tend to rely on the local neighbourhood information of the patch data". Isn't this exactly the selling point for patch-based CNNs, that they can indeed make use of the local spatial context information?
Lines 141 f. (as well as Discussion and Conclusion): How is increasing the classification accuracy in randomly shuffled data an advantage?
And overall, back to the major conceptual issue: As the defining property of patch-based CNNs is to make use of spatial local context information, isn't it trivial to show that they perform worse on randomly shuffled data and especially, that they, as proportion of random shuffling data increases, cannot extract more abundant local context information?
Line 128 ("we [...] design a novel strategy to reconstruct the original data set") and Methodology 2.1: How is random shuffling a reconstruction technique? Is reconstructed data = randomly shuffled data? Also, in Figure 1, the data cubes might be misinterpreted. Why do the original and reconstructed cube have different orientation and (seemingly) also a different number of channels? Use subfigures to visualize a different points of view, like the "planar effect", if necessary.
Methods 2.2 about the SKDN network contains too much background information about how to implement a Siamese network for classification and knowledge destillation in general, which is not directly relevant for the actual method proposed and therefore should be deleted. Further, this part contains very long sentences, minor faults and is missing some articles (e.g., line 210 ff.), and therefore is hard to understand. Some editing of the content as well as the English language is required.
Experiments 3.2 where the newly propsed model is applied to four real-world data sets is basically the only experimental part of interest and then, for the relevant ratio of 0%, in most cases, the improvement is only very small to negligible.
Further, why is, e.g., the training ratio chosen differently for the four different data sets (line 274 f.)?
In general, the experimental results are described in too much detail (e.g., giving too much examples and numbers in the text), which makes the Experiments part lengthy and also rather hard to read.
The English language is fine, except for section 2.2. This part contains very long sentences, minor faults and is missing some articles (e.g., line 212 ff.). Some editing of the English language is required for the whole section.
Reviewer 2 Report
This manuscript introduces SKDN, a new network architecture, which incorporates a novel sub-branch called RFL. Furthermore, a random shuffling strategy is employed to reconstruct the HSI dataset. Experimental evaluations on various datasets reveal that the proposed approach significantly enhances the extraction of local contextual information, resulting in improved HSI classification performance compared to the original model. The following comments would help to improve the quality of the manuscript:
(a) Please provide more specific descriptions of the proposed model SKDN. In addition, I’m curious why SKDN doesn’t have a full name?
(b) Please explain the specific meaning of L_rec (Eq. 1), PCA (Fig. 3), and RFL sub-branch and correspond to the manuscript narrative.
(c) Please analyze the results in Table 2-7 rather than simply list them.
(d) There are some typos in the manuscript, for example, on line 37, ‘CNNs’ should be written in its full name as it appears for the first time. Additionally, references 2, 3, 4, 14, and 21 are not formatted correctly. Please carefully review the references to ensure their consistency and correctness.
Moderate editing of English language required.
Reviewer 3 Report
The authors propose a new sub-branch called RFL, which is integrated into the original network to construct a new network called SKDN. The authors also introduce a random shuffling strategy to reconstruct the HSI dataset. The experiments on several datasets show that the proposed method can effectively extract more abundant local contextual information for HSI classification, and the classification performance of the improved model is better than that of the original model. However, the manuscript does not contain highlights that would attract readers, and some parts would need to be carefully revised.
1. Introduction section is difficult to follow and confusingly structured, and please clarify the motivation, research question and contribution of the paper.
2. Add more description of Fig. 1, i.e., how the stated Randomly shuffling the pixels works.
3. Evaluation metrics (OA, AA and Kappa) should be specified in Table 2 and 3.
4. Experimental sections should include ablation studies, such as the effect of removing RFL sub-branches on model performance.
5. It should compare with the SOTA results in 2023 so as to demonstrate the advancement of the proposed methods.
The authors should try their best to improve the quality of English.
Round 2
Reviewer 1 Report
The authors did their best to respond to all of my comments and to make appropriate changes to their paper. Overall, I am satisfied with the result. However, the English language still needs to be improved.
The English language still needs to be improved.
Reviewer 2 Report
This manuscript could be accepted in present form.
The quality of English should be improved by the native speaker.
Reviewer 3 Report
The authors solve all my concerns, and please accept this article.
Minor editing of English language required.